# Worldwide Traceability of Antibiotic Residues from Livestock in Wastewater and Soil: A Systematic Review

**DOI:** 10.3390/ani12010060

**Published:** 2021-12-28

**Authors:** Lizbeth E. Robles-Jimenez, Edgar Aranda-Aguirre, Octavio A. Castelan-Ortega, Beatriz S. Shettino-Bermudez, Rutilio Ortiz-Salinas, Marta Miranda, Xunde Li, Juan C. Angeles-Hernandez, Einar Vargas-Bello-Pérez, Manuel Gonzalez-Ronquillo

**Affiliations:** 1Departamento de Nutricion Animal, Facultad de Medicina Veterinaria y Zootecnia, Instituto Literario 100 Ote, Universidad Autonoma del Estado de Mexico, Toluca 50000, Mexico; lizroblez@hotmail.com (L.E.R.-J.); edgarfmvzaa@gmail.com (E.A.-A.); oaco2002@yahoo.com.mx (O.A.C.-O.); 2Laboratorio de Análisis Instrumental, Departamento de Producción Agrícola y Animal, Universidad Autónoma Metropolitana Xochimilco, Calz, Hueso 1100, Villa Quietud, Coyoacan 04960, Mexico; schettinob@yahoo.com.mx (B.S.S.-B.); rortizs@correo.xoc.uam.mx (R.O.-S.); 3Departamento de Anatomia, Produccion Animal y Ciencias Clinicas Veterinarias, Facultad de Veterinaria, Universidad de Santiago de Compostela, 27002 Lugo, Spain; marta.miranda@usc.es; 4Western Institute for Food Safety and Security, School of Veterinary Medicine, University of California Davis, Davis, CA 95616, USA; xdli@ucdavis.edu; 5Instituto de Ciencias Agropecuarias, Universidad Autónoma del Estado de Hidalgo, Av. Universidad km 1, Tulancingo de Bravo 43600, Mexico; juan_angeles@uaeh.edu.mx; 6Department of Veterinary and Animal Sciences, Faculty of Health and Medical Sciences, University of Copenhagen, Grønnegårdsvej 3, DK-1870 Frederiksberg C, Denmark; 7Sciences Department, Faculty of Humanities, Social and Health Sciences, Universidad de Magallanes, Avenida Bulnes, Punta Arenas 01855, Chile

**Keywords:** antibiotic residues, livestock, wastewater, antimicrobial resistance, soil residues

## Abstract

**Simple Summary:**

This work focuses on reviewing research works on antibiotic residues, evaluating antibiotics used in livestock production and their excretion in animal products and in environmental matrices such as water and soil worldwide, according to each of the variables used such as antibiotic family, name, concentration (% and mg/kg or ppm), country, and continent where the residue was found. The main antibiotics used worldwide and in animal production are sulfonamides, tetracyclines, quinolones, penicillin, and cephalosporins.

**Abstract:**

The use of antibiotics in animal production are widely used for disease treatment, health protection, and as growth promoters. Common antibiotics used in veterinary medicine are excreted and eliminated through the sewage system, contaminating water and soil with negative effects on agricultural activities. This systematic review focuses on the trend of research works on antibiotic residues, evaluating antibiotics used in livestock production and their excretion in animal products and in environmental matrices such as water and soil. Our database was composed of 165 articles, reporting the concentration of antibiotic residues found in the environment, livestock (cow, sheep, pig, horse, chicken, rabbit, goat), aquatic and terrestrial animal tissues, animal products (milk and eggs), wastewater, and soil. The documents were obtained from Asia, Africa, North America, South America, Europe, and Oceania. A descriptive analysis of antibiotic residues found worldwide was analyzed according to each of the variables used such as antibiotic family, name, concentration (% and mg/kg or ppm), and country and continent where the residue was found. The descriptive analysis was carried out using the “describe” function of psych package and pirate plots were drawn. According to our study, the main antibiotics used worldwide in animal production are sulfonamides, tetracyclines, quinolones, penicillin, and cephalosporins. At present, despite the trends of increased regulations on the use of antibiotics worldwide, antibiotics are still utilized in food animal production, and are present in water and soil, then, there is still the misuse of antibiotics in many countries. We need to become aware that antibiotic contamination is a global problem, and we are challenged to reduce and improve their use.

## 1. Introduction

The absorption of antibiotics in animals after administration is often poor and a significant proportion of 70–90% may be excreted unmetabolized [1], and these residues remain unchanged in the environment [2,3]. The use of antibiotics in animal production has been increasing and they are widely used for disease treatment, health protection [4,5,6], and as growth promoters [7,8]. However, the family and active ingredient of the antibiotic vary with animal species (i.e., oxytetracycline, chlortetracycline, and tylosin are used in pigs in weaning and finishing; beta-lactam and tetracyclines in dairy cows; florfenicol and spectinomycin in calves) and the facet of production in which it is used [9,10]. All antibiotics used in veterinary medicine [11,12] are excreted and disposed over the sewage system, and in some cases, to sewage treatment plants [13]. This situation has polluting effects on water and soil with negative effects on agricultural activities, for example, it has been mentioned that in the soil, antibiotic residues are responsible for anoxic denitrification since they affect bacterial communities responsible for this process [14]. 

Antibiotic residues in food of animal origin also pose health risks such as bacterial resistance, toxicity, hypersensitivity reactions, cancer, and teratogenicity [15]. In May 2015, the 68th World Health Assembly recognized the importance of antimicrobial resistance and adopted a plan to reduce the unnecessary use of antimicrobials in humans and animals [16,17]. Thus, since 2006, member countries of the European Union [18], and since 2011, New Zealand and the Republic of Korea have banned the use of antibiotics as growth promoters [19]. Other countries such as Australia, Canada, Japan, and the United States have applied policies and regulations to ensure that they are only used by licensed veterinarians [20,21,22]. Large meat producing countries such as Argentina, Brazil, China, India, Indonesia, the Philippines, Russia, and South Africa have not banned the use of antibiotics as growth promoters [23]. The greatest uncertainty about antibiotic use in livestock is found in low-income countries due to the lack of information on the use of antibiotics [16]. The differences that exist in the use of antibiotics both by species and by country due to the policies implemented worldwide make quantification very difficult (i.e., only 42 countries in the world have a system for collecting data on the use of antimicrobials in livestock) [24].

Antibiotic use is estimated to increase by 67% by 2030, with China, Brazil, India, South Africa, and Russia being the main consuming countries [25]. Therefore, this review focuses on the trend of research work on antibiotic residues found in environmental samples such as water, soil, and livestock products, aquatic and terrestrial animal tissues, and animal products (milk and eggs).

## 2. Materials and Methods

### 2.1. Search Strategy and Selection Criteria

The information search focused on studies reporting veterinary antibiotic residues found in the environment, wastewater, soil, and their bioaccumulation in animal tissues and products worldwide. For this purpose, a database of publications specifying antibiotic residues worldwide was created and the articles used covered the years 2000 to 2019. The publications were obtained from different databases such as ScienceDirect 2021, Scopus, Di-alnet, SciELO, Science Research, PubMEd, Redalyc, and Google Scholar.

The search string with the particular topic was supported by Boolean operators (“and”, “or”). All search terms within a string were checked for a “title, abstract, and keyword”. The keywords used were antibiotic residues in the environment (wastewater and soil), traceability, animal husbandry, animal species (cow, sheep, pig, horse, chicken, rabbit, goat), aquatic and terrestrial animal tissues, animal products (milk and eggs), bioaccumulation in animal tissues, and antibiotic concentrations (% and mg/kg). The search for information was carried out by continent.

The Council Directive 96/23/EC, Annex 1 [26], classifies veterinary medicinal products and substances with anabolic effects used in animal feed into two groups: Group A and Group B. Group A contains substances that have anabolic effects such as stilbenes (diethylstilbestrol), steroids, androgens (trenbolone acetate), gestagens (melengestrol acetate), oestrogens (17-beta estradiol), resorcyclic acid lactones (zeranol), beta agonist (clenbuterol), and nitrofurans. Group B contains all veterinary medicinal products (e.g., sulfonamides, quinolones).

The ranking order of antibiotic families based on their occurrence (%) is shown in Table 1 [27]; antibiotics highlighted in bold letters represent their family and the most widely used antibiotics, respectively.

### 2.2. Data Extraction and Analysis

Our database was composed of 165 articles (Figure 1) reporting the concentration of antibiotic residues found in the environment, livestock (cow, sheep, pig, horse, chicken, rabbit, goat), aquatic and terrestrial animal tissues, animal products (milk and eggs), wastewater, and soil (Appendix A). The study focused on assessing the presence of antibiotics in wastewater management systems, which are mainly used in semi-urban, rural, and remote areas as well as animal farms, where the installation of a centralized sewage system is not feasible, and many of these wastes seep through groundwater or simply remain in the soil, hence the importance of showing that antibiotic residues exist in water and soil. The documents were obtained from Asia, Africa, North America, South America, Europe, and Oceania (Table 2). A descriptive analysis of antibiotic residues found worldwide was analyzed according to each of the variables used such as antibiotic family, name, concentration (% and mg/kg or ppm), and country and continent where the residue was found. The descriptive analysis was carried out using the “describe” function of psych package [28] and pirate plots were drawn using the Yarrr package [29].

## 3. Results

### 3.1. Veterinary Antibiotics as Pollutants in Different Continents

From all antibiotics produced worldwide in 2015, two-thirds (65,000 tones) were used for animal husbandry. The highest consumption of antibiotics in livestock was in China (>15,000 tons), followed by the USA with 9000 tons, while France and Canada reported a consumption of approximately 2000 tons [30].

According to the antibiotic management situation, the WHO has tried to create an observational and ecological database to define which antimicrobials are medically important. Recommendations and web pages have been derived for consultation as defined by the Guideline Development Group (GDG) [31].

The residue levels of antibiotics based on continent showed a marked variability among antimicrobial families. The antimicrobial with the highest concentration in Asia was cephalosporins (450 ± 353.55 ppm), followed by fluroquinolone (129.44 ± 509.81 ppm). The tetracyclines were the antibiotic family with highest residual concentration in Africa (176.74 ± 930.75 ppm) and North America (106.11 ± 146.86 ppm). In South America, the family of antibiotics that depicted the highest level of residues was fluroquinolones (726.91 ± 1437.29 ppm), followed by macrolides (407 ± 574.17 ppm). In Europe, the largest concentration of residues was shown by ß-lactam (509.2 ± 1220.29 ppm), followed by nitroimidazole (250 ppm). The studies developed in Oceania only reported antimicrobial resistance of aminoglycoside (8.41 ± 15.74 ppm) and fluoroquinolones (1.9 ± 2.68 ppm) (Figure 2).

### 3.2. Residues of Veterinary Antibiotics in Animal Products and Derivatives

According to the main animal products reported in the analyzed studies, the largest concentration of residues was found in chicken (341.44 ± 1025.76 ppm), shrimp (259.02 ± 691.38 ppm), and cow’s milk (100.65 ± 318.41 ppm) (Figure 3). Studies that reported microbial residues of pork meat showed the largest concentration in nitroimidazole (15 ppm), tetracyclines (10.25 ± 17.23), and aminoglycoside (4.54 ± 6.15 ppm). Aminoglycosides were the antimicrobials that had the highest levels of residues in beef meat (2.1 ± 2.34 ppm). In cow’s milk, the cephalosporins and macrolides were the families of antibiotics that showed the highest levels of residues. However, these antibiotic families had a reduced number of reports. On the other hand, there were several studies that reported residues of tetracyclines in cow’s milk with a high variability (132.36 ± 480.19 ppm). For example, a study [32] reported concentrations of tetracycline residues as high as 1800 ppm (Figure 4). Studies [33,34] that have reported residues of antimicrobial in sheep meat only referred to concentrations of fluoroquinolones (0.73 ± 0.69 ppm), tetracyclines (0.5 ± 0.54 ppm), and ß-lactam (0.04 ± 0.04 ppm). 

Tetracyclines were the antibiotics that had more information and showed the largest concentration of residues found in eggs (70.29 ± 139.84 ppm). Although the fluoroquinolones showed the highest average value of residues of antimicrobial in chicken (981.83 ± 1486.48 ppm), the tetracyclines showed the concentrations in this animal products, which were above 5000 ppm with an average of 438 ± 1441.29 ppm. Studies focusing on the evaluation of antimicrobial residues in fish reported the highest concentration of nitroimidazole (250 ppm; *n* = 1), however, the largest number of studies focused on tetracyclines, which showed the second highest concentration of residues as well (51.75 ± 49.67 ppm; *n* = 8). Fluroquinolones (612.17 ± 1134.83 ppm) and sulfonamides (205.74 ± 409.83 ppm) were the antibiotics that showed the highest concentration of residues in shrimp (Figure 4).

### 3.3. Impact of Antibiotic Residues in Water and Soil

Wastewater management is defined as the collection, treatment, and reuse of wastewater at or near the point of waste generation, in this sense, the present study focused on assessing the presence of antibiotics in wastewater management systems, which are mainly used in semi-urban, rural and remote areas as well as from animal farms, where the installation of a centralized sewerage system is not feasible, and many of these wastes seep through groundwater or simply remain on the soil, hence the importance of determining the waste of antibiotic residues. 

In addition, the present study shows that fluoroquinolones were the antibiotics that showed the largest concentration in soil (32.34 ± 97.52) and water (325.0 ± 471.69 ppm) (Figure 5 and Figure 6), fluoroquinolones may develop disabling and potentially permanent side effects of the tendons, muscles, joints, nerves, and central nervous system.

## 4. Discussion

### 4.1. Veterinary Antibiotics as Pollutants in Different Continents

The largest user of veterinary antibiotics worldwide are China (45%), Brazil (7.9%), USA (7.0%), Thailand (4.2%), India (2.2%), Iran (1.9%), Spain (1.9%), Russia (1.8%), Mexico (1.7%), and Argentina (1.5%) [27]. It is important to note that there is a particular distribution of antibiotic use by geographical area depending on their policies, economic/market conditions, and dietary habits [35].

The regulation of antibiotics in animal feed (for growth promotion or therapeutic use) is a priority for the Asian region, and policies for their prohibition have been developed, but few countries have the capacity to guarantee their application [36]. The Chinese government has implemented policies to control the use of antibiotics: the use of any antibiotics included in the prohibited list is banned, the use of antibiotics during the waiting period is prohibited, the purchase of antibiotics without veterinary prescription is prohibited, the use of medically important antibiotics in food animals is prohibited, however, they are still purchased without prescription and not monitored at most farms [37]. In Russia, farmers can use antibiotics without any restrictions, while some feed antibiotics are subject to state control [38]. Iran is one of the countries where the sale of antibiotics is not controlled, and no prescription is needed to purchase the drugs. This is due to the lack of national action plans to try to control this problem, low awareness of farm producers, fragmented information systems due to political problems, so that monitoring and surveillance is irregular [39]. In Thailand, surveillance of antibiotic use is insufficient due to gaps in human resources, particularly for smallholder farmers, and there is little information from farmers on the resistance that can be caused by antibiotics misuse [40].

In India, in the last decade, changes in the population diet generated by the improved standards of living, has led to a demand for animal protein with consequent intensification of pig, poultry, and fish farming [41], which has had a significant effect on the use of antibiotics such tetracyclines, penicillins, and sulfonamides. However, the highest concentrations of antibiotics were found for cephalosporins and fluroquinolones (Figure 2). This higher concentration may be related to the fact that these antibiotics are frequently used in swine production for respiratory problems as well as in aquaculture, and Asia allocates most of the veterinary antibiotics to swine production [42]. This is mainly due to the fact that the use of antibiotics in the livestock sector in Asia presents the weakness or the lack of regulations, adequate policies, and the implementation of quality standards, causing the emergence and spread of antibiotic resistance today [36,43].

In Africa, the main antibiotics used for livestock farming are tetracyclines, fluoroquinolones, and β-lactams/aminoglycosides (33.6%, 26.5%, and 20.4%, respectively) [44]. In our study, the above-mentioned can be confirmed since tetracyclines and fluoroquinolones are the most frequently found antibiotics in animal products (Figure 2). Factors influencing the excessive use of antibiotics in Africa are the accessibility to purchase drugs as well as the types of animals kept (e.g., poultry) and the farming system of the animals (e.g., intensive); however, farmers had limited experience on their use for animal production [6]. Weak national surveillance systems, lack of coordination between national authorities and the private sector, lack of human resources, and insufficient regulatory standards are other reasons for the increase in the overuse of antibiotics in Africa [45].

A study in 30 European countries found that the main antibiotics sold were tetracyclines (32.8%), penicillins (25.0%), and sulfonamides (11.8%) [46]. Together, these three classes accounted for 69.6% of total sales [47], however, the highest concentrations of antibiotics were found for ß-lactam (509.2 ± 1220.29 ppm), followed by nitroimidazole (250 ppm). Although the use of antibiotics in the EU is more restricted and monitored, some member countries still use large quantities, for example, countries such as Spain and Italy consume significantly more drugs than Northern members of the EU [48]. This can be seen in our study since in Table 2, the highest number of articles reported with antibiotic residues in Europe was from Spain, with six articles.

When talking about meat producers, one must mention the world’s leading meat producers, North and South America, which are also among the main consumers of veterinary antibiotics [45,46]. For example, the United States uses 24.6 million pounds of antimicrobials for non-therapeutic purposes in chickens, cattle, and pigs, the most commonly used being tetracyclines, penicillins, fluroquinolones, and sulfonamides [49] and Brazil, being the fourth largest pork producer in the world, uses sulfonamides [50]. However, although government agencies are trying to regularize the use of antibiotics, producers are not willing to stop using them, as they consider that it would be impossible to sustain current market demands without the use of antimicrobials [50]. In our study, we can confirm this since most of the articles found on antibiotic residues were from the USA and Mexico (Table 2). Although Brazil is one of the largest meat producers, only one article was found, which may be due to the lack of published reports and complex political barriers [48,49].

Although most countries have joined in the program of non-misuse of antibiotics in food animals, publication of reports, complex political, economic, and social barriers still limit the quality of data on this issue [48,49]. Data on their use of antibiotics is readily available from countries that export a significant portion of their animal production than those countries where the majority of their production is destined for the domestic market [51]. All of the above were limitations to obtaining more articles needed to reinforce the obtained information, and until now, the consumption of antibiotics at present is still indiscriminant.

### 4.2. Residues of Veterinary Antibiotics in Animal Products and Derivatives

Antibiotics have been used for three main purposes in animals: therapeutic use against infectious diseases; prophylactic use for the prevention of these diseases; and as growth promoters to improve feed utilization and animal production [52]. The growth-promoting effects of antibiotics were discovered in the 1940s, when chickens were fed tetracycline fermentation byproducts. In this case, chickens exhibited higher growth rates compared to those that were not fed feed containing the byproducts [53]. Since then, the use of growth promoters has expanded to include a wide range of antibiotics that are applied to various species. The use of antibiotics in animal production offers proven benefits in animal health and production as well as a reduction in foodborne pathogens [54]. However, due to the increasing concern of antibiotic resistance, the use of antibiotics as growth promoters in livestock industry has been recently banned in the U.S. [55] and Europe (Regulation (EU) 2019/6 [56]).

Overuse or lack of control in antibiotic stewardship results in high antibiotic deposition in the animal and excretion into the environment [57]. Olatoye and Ehinmoro [58] detected approximately 54% of oxytetracycline residues in livestock. In Egypt, in fresh chicken meat and liver samples, 44% of the samples contained tetracycline residues, ranging from 38 to 52%, and the corresponding contamination residues ranged from 103 to 8148 μg/kg, which were above the Codex maximum residue limit (200 μg/kg and 600 μg/kg for chicken meat and liver, respectively, expressed as the sum of the tetracycline group) [59]. European Union (EU) No. 37/2010 has stipulated that the maximum residue limit for all synthetic antimicrobials such as sulfonamides is 100 μg/kg in edible animal tissue [60]. According to the U.S. FDA reports, tetracyclines show the highest level of drug application, followed by sulfonamides and aminoglycosides [61]. 

It has been mentioned that pigs are the main consumers of antimicrobials and are expected to use 45% of antimicrobials for animal production from 2017 to 2030, with cattle using 22% and chickens 33% of antimicrobials. On average, pigs consume 193 mg/PCU (stock correction unit), cattle consume 42 mg/PCU of antimicrobials, and chickens consume 68 mg/PCU of antimicrobials [51]. However, although antibiotics are mostly used in the pig industry, chicken meat and cow’s milk had the highest concentrations of antibiotics, which disagrees with results found in the present work (Figure 3).

In pigs, depending on their production stage, the use of antibiotics changes, for example, oral antibiotics are used as routine prophylaxis for fattening and medicated feed as growth promoters for weaners. However, in all phases, the most widely used are aminopenicillin, tetracyclines, trimethoprim-sulfonamides, tylosin, and colistin [10]. This was partly confirmed in the present study, since the main antibiotic residues found in pork were sulfonamides, tetracyclines, and quinolones (Figure 4), which coincides with Ramatla et al. [62].

In the present study, we found that the main antibiotic residues found in beef were sulfonamides (30%), which coincides with Ramatla et al. [62] and Treiber and Beranek-Knaue [63]. This may be because in veterinary practice, due to their broad spectrum of activity and low cost as well as to promote the growth of animals, sulfonamides are widely used in food animal species including cattle, swine, poultry, and aquaculture [64,65]. However, the highest concentrations were for aminoglycosides (2.1 ± 2.34 ppm). 

Antibiotics used to combat mastitis-causing pathogens are the most common purpose of use in the dairy industry [52], However, cephalosporins, macrolides, and tetracyclines had the highest concentrations of residues in milk (Figure 4, Cow milk). LeBlanc et al. [66] conducted an extensive review of antimicrobial resistance in adult dairy cows and concluded that the use of antibiotics in the dairy industry for treatment and prevention contributes to increased antimicrobial resistance.

Pereira et al. [67] evaluated the effect of antimicrobial use on drug resistance in fecal *E. coli* isolated from pre-weaned dairy calves and found that isolates from calves treated with enrofloxacin were more likely to be resistant to fluoroquinolones. This example indicates an important concern, as these antimicrobial agents are essential in human medicine for treatment against *Salmonella spp., Campylobacter spp.* and *Shigella spp*. There are also other reports of antibiotic and anthelmintic residues in dairy products [67,68] and tetracyclines are one of the main antibiotics found in dairy products according to the data collected in this article.

The use of antibiotics as growth promoters in broilers is intended to modify the intestinal flora, thereby improving feed absorption to increase muscle mass by 15–20% in a short time. In addition, antibiotics are commonly used in high concentrations in overcrowded poultry [69]. This can be seen in our work, as the most common antibiotics found in chickens were fluoroquinolone, tetracycline, and aminoglycoside. It is important to note that although many antibiotics are used more frequently in animal production, the residues found in products intended for human consumption of animal origin were different from those found in the present work, which may be influenced because the persistence of antibiotic residues is affected by the animal species from which they come, their respective diet, and intestinal microbiota, in addition to the fact that some antibiotics degrade more rapidly (hours) [70] with respect to others, and the time that the antibiotic was administered until the animals were slaughtered [71].

The marketing of antibiotics for veterinary use is often not adequately restricted as there are cases of empirical supply without prescription. In addition, there is a lack of adequate registration in the control of medicines by the health authorities as well as periodic monitoring of infectious agents with zoonotic potential in backyard, semi-intensive, and intensive production sites. Good production practices are not considered by all producers from prevention, treatment, and slaughter due to the lack of maintenance of facilities [72]. In the present study, we observed that many countries used large amounts of antibiotics that are found in animal products of any animal species in different concentrations, which confirms the above-mentioned. 

Most animal production focuses on poultry, pigs, and cattle, but aquaculture facilities, where there is a high use of antibiotics, must also be taken into account [73]. Oxytetracycline, florfenicol, sarafloxacin, and sulfonamides are widely used in aquaculture and are therefore detected in aquatic samples [74], where, as shown in Figure 3 and Figure 4, quinolones, tetracyclines, and sulfonamides are the most commonly used in fish and shrimp, which coincides with Guidi et al. [75]. 

Open water aquaculture also employs antibiotics to prevent disease and promote the growth of target species leading to evolution and a wide dispersal of resistance agents (ARBs and ARGs) in water and their possible deposition in sediments. This is of particular interest in systems where there is a connection of animal production and aquaculture as there is a closer relationship of resistance gene transfer between the systems involved [76]. 

As above-mentioned, there is evidence of some negative consequences of antibiotic use in animal production, especially in bacterial resistance, particularly zoonotic microorganisms such as *Salmonella spp, Campylobacter spp*, *Enterococcus faecalis*, and others [77]. Therefore, records on the effects of antibiotics in veterinary use is needed for the development of national and international policies related to the issue of microbial resistance.

### 4.3. Impact of Antibiotic Residues in Water and Soil

With the aforementioned background such as the inappropriate use of antibiotics in animal production, antibiotic residues are excreted in feces or urine and present in environmental matrices such as soil, water, and vegetation, which may cause risks to human health [78], hence the latest trends in antimicrobial resistance. For example, in the United States (U.S.), 70% of antibiotics were used in animal production, which is eight times the amount used in human medicine [79]; in 2013, U.S. livestock producers purchased 14,900 tons of antimicrobials [80], which represents a large amount of antibiotics potentially disposed of into the environment (aquatic waste and soil). Van Boeckel et al. [25] estimated that between 2010 and 2030, the use of antibiotics in food animal production will increase by 67%, from 63,151 ± 1560 tons to 105,596 ± 3605 tons, which could increase the issue of antimicrobial resistance in future generations.

Many of the antibiotics and hormones administered to animals cannot be fully absorbed or metabolized in the body and are excreted directly into the wastewater system [81,82,83]. Consequently, wastewater systems are an important pathway for the removal and distribution of antibiotics [84]. Many antibiotics present in wastewater are removed and transported to sewage sludge during sewage treatment [85], indicating that sludge can serve as an important reservoir for antibiotics (Figure 6). Furthermore, it has been recognized that sludge disposal such as agricultural application and landfilling can potentially release antibiotics into the environment and may pose potential risks to animal and human health ecosystems [86,87].

Triclosan, sulfonamides, and trimethoprim are the most frequently found antibiotics in soil [71], however, the highest concentrations found were in fluoroquinolones (32.34 ± 97.52) (Figure 6). Quinolones, sulfonamides, and trimethoprim have been the most frequently found antibiotics in water, which exceed 1 μg/L in environmental samples [88] (Figure 5; fluoroquinolones 325.0 ± 471.69 ppm). The persistence of these antibiotics in the environment may be due to their degradation time, since some antibiotics such as penicillin degrade easily within hours or a few days, while other antibiotics such as macrolides (i.e., tylosin), fluoroquinolones, and tetracyclines can persist for several months or even years (Figure 6, Soil) [89].

It has been shown through molecular markers that the main cause of antibiotic residues in the environment is due to livestock feces as doses much higher than those prescribed are applied to the human population, as demonstrated in animal excrement in the United States, where values exceeded 100 times more than those found in wastewater of anthropogenic origin [90]. 

It is important to control the doses used in livestock production to avoid pathogen resistance to antibiotics and water contamination as wastewater in some areas is used as an alternative for the irrigation of agricultural fields [45,46]. 

Animal production systems have separate drains from the municipal ones, and in many cases, the waste is concentrated in waste lagoons where antibiotics accumulate. In these lagoons, sludge or water is obtained and applied to nearby land, so there is close contact with soil microbiota, or in rainy seasons, there is runoff or filtration of these compounds that reach aquatic bodies or groundwater [91]. Therefore, there is environmental pressure on microorganisms in soil and water when antibiotic contamination is present in the environment, forcing a selection on the reduction in the diversity and composition of the microbial community. Appreciating that antibiotic exposure tends to favor an increase in Gram-negative bacteria compared to Gram-positive bacteria, this will result in the disruption or loss of bacteria that play key ecological roles such as in the decomposition of matter [92,93].

## 5. Conclusions

In recent years, it has been demonstrated that antibiotic residues are present worldwide in wastewater, soil, and animal production, the most common being sulfonamides, tetracyclines, quinolones, penicillins, and cephalosporins. New international policies have limited their use as therapeutics, restricting their use as growth promoters in animal production. Intensive livestock production must change, as it would be impossible to sustain current market demands without the use of antimicrobials or friendlier alternatives, with a future decrease in antimicrobial resistance, so we are challenged to reduce their use.

## Figures and Tables

**Figure 1 animals-12-00060-f001:**
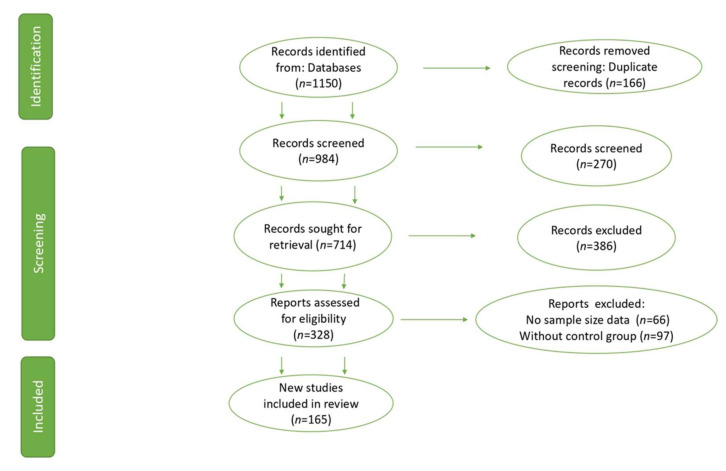
PRISMA study flow diagram of the systematic review from the initial search and screening to the final selection of publications to be included in the systematic review.

**Figure 2 animals-12-00060-f002:**
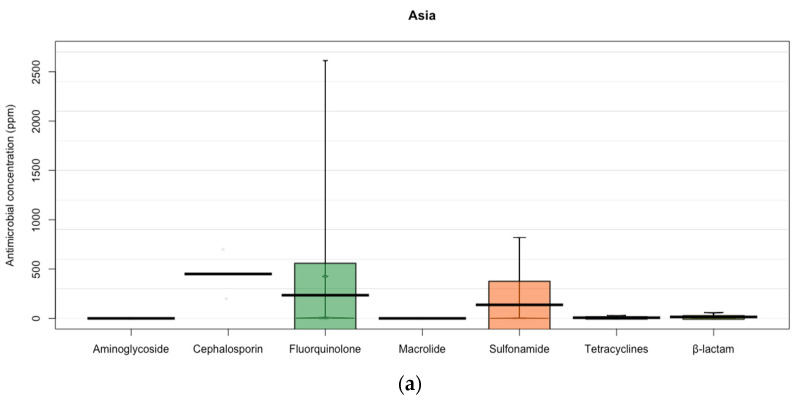
Antibiotics (ppm) used in Asia, Africa, North America, South America, Europe, and Oceania. (**a**) Antibiotics (ppm) used in Asia. Points represent the raw data; bar/line is the descriptive statistic (mean); bean is the smoothed density curve showing the full data distribution; and brackets represent the confidence intervals. (**b**) Antibiotics (ppm) used in Africa. Points represent the raw data; bar/line is the descriptive statistic (mean); bean is the smoothed density curve showing the full data distribution; and brackets represent the confidence intervals. (**c**) Antibiotics (ppm) used in North America. Points represent the raw data; bar/line is the descriptive statistic (mean); bean is the smoothed density curve showing the full data distribution; and brackets represent the confidence intervals. (**d**) Antibiotics (ppm) used in South America. Points represent the raw data; bar/line is the descriptive statistic (mean); bean is the smoothed density curve showing the full data distribution; and brackets represent the confidence intervals. (**e**) Antibiotics (ppm) used in Europe. Points represent the raw data; bar/line is the descriptive statistic (mean); bean is the smoothed density curve showing the full data distribution; and brackets represent the confidence intervals. (**f**) Antibiotics (ppm) used in Oceania. Points represent the raw data; bar/line is the descriptive statistic (mean); bean is the smoothed density curve showing the full data distribution; and brackets represent the confidence intervals.

**Figure 3 animals-12-00060-f003:**
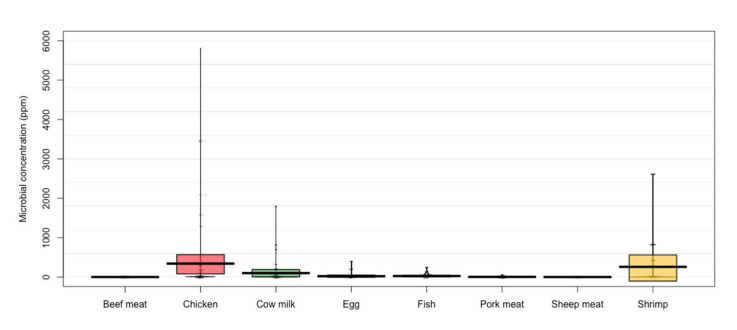
Presence of antibiotics in animal products. Points represent the raw data; bar/line is the descriptive statistic (mean); bean is the smoothed density curve showing the full data distribution; and brackets represent the confidence intervals.

**Figure 4 animals-12-00060-f004:**
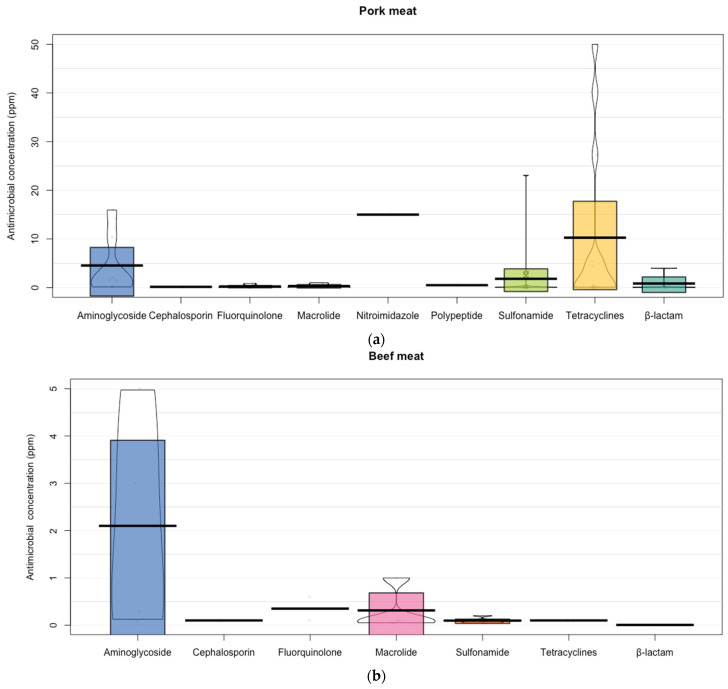
Presence of antibiotics in animal products according with the antimicrobial family. (**a**) Presence of antibiotics in Pork meat according with the antimicrobial family. Points represent the raw data; bar/line is the descriptive statistic (mean); bean is the smoothed density curve showing the full data distribution; and brackets represent the confidence intervals. (**b**) Presence of antibiotics in Beef meat according with the antimicrobial family. Points represent the raw data; bar/line is the descriptive statistic (mean); bean is the smoothed density curve showing the full data distribution; and brackets represent the confidence intervals. (**c**) Presence of antibiotics in cow’s milk according with the antimicrobial family. Points represent the raw data; bar/line is the descriptive statistic (mean); bean is the smoothed density curve showing the full data distribution; and brackets represent the confidence intervals. (**d**) Presence of antibiotics in Sheep meat according with the antimicrobial family. Points represent the raw data; bar/line is the descriptive statistic (mean); bean is the smoothed density curve showing the full data distribution; and brackets represent the confidence intervals. (**e**) Presence of antibiotics in Egg according with the antimicrobial family. Points represent the raw data; bar/line is the descriptive statistic (mean); bean is the smoothed density curve showing the full data distribution; and brackets represent the confidence intervals. (**f**) Presence of antibiotics in Chicken according with the antimicrobial family. Points represent the raw data; bar/line is the descriptive statistic (mean); bean is the smoothed density curve showing the full data distribution; and brackets represent the confidence intervals. (**g**) Presence of antibiotics in Fish according with the antimicrobial family. Points represent the raw data; bar/line is the descriptive statistic (mean); bean is the smoothed density curve showing the full data distribution; and brackets represent the confidence intervals. (**h**) Presence of antibiotics in Shrimp according with the antimicrobial family. Points represent the raw data; bar/line is the descriptive statistic (mean); bean is the smoothed density curve showing the full data distribution; and brackets represent the confidence intervals.

**Figure 5 animals-12-00060-f005:**
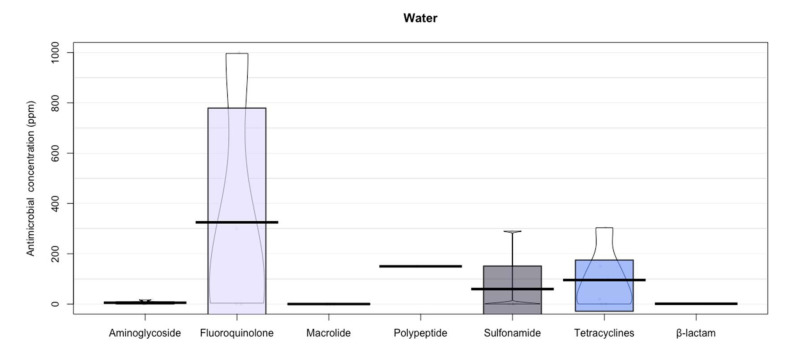
Antibiotic residues in water. Points represent the raw data; bar/line is the descriptive statistic (mean); bean is the smoothed density curve showing the full data distribution; and brackets represent the confidence intervals.

**Figure 6 animals-12-00060-f006:**
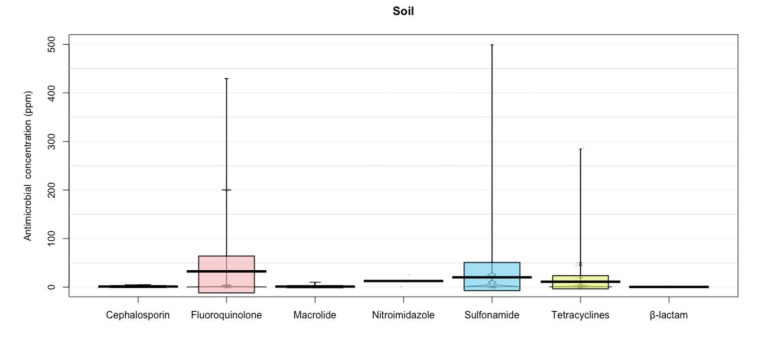
Antibiotic residues in soil. Points represent the raw data; bar/line is the descriptive statistic (mean); bean is the smoothed density curve showing the full data distribution; and brackets represent the confidence intervals.

**Table 1 animals-12-00060-t001:** Categorization per family of veterinary antibiotics for food-producing animals, representing their occurrence (%).

Antibiotics									
Penicillins 87.1%	Tetracyclines 87.1%	Aminoglycosides 77.1%	Macrolides 77.1%	Sulfonamides 70%	Quinolones 68.6%	Polypeptides 64.3%	Cephalosporins 58.6%	Phenicols 51.4%	Lincosamides 51.4%
*Natural Penicillins* Benzylpenicillin Penethamate hydroxide Penicillin procaine *Amdinopenicillins* Mecillinam *Aminopenicillins* Amoxicillin Ampicillin Hetacillin *Aminopenicillin plus Betalactamase inhibitor* Amoxicillin_Clavulanic Acid *Carboxypenicillins* Ticarcillin Tobicillin *Ureido**Penicillin* Aspoxicillin *Phenoxypenicillins* Phenoxymethylpenicillin Phenethicillin *Antistaphylococcal Penicillins* Cloxacillin Dicloxacillin Nafcillin Oxacillin	Chlortetracycline Doxycycline Oxytetracycline Tetracycline	*Aminocyclitol* Spectinomycin *Aminoglycosides* Streptomycin Dihydrostreptomycin Framycetin Kanamycin Neomycin Paromomycin Apramycin Gentamicin Tobramycin Amikacin	*Azalide* Tulathromycin *Macrolides**C14* Erythromycin *Macrolides C16* Josamycin Kitasamycin Spiramycin Tilmicosin Tylosin Mirosamycin Terdecamycin	Sulfachlorpyridazine Sulfadiazine Sulfadimerazin Sulfadimethoxine Sulfadimidine Sulfadoxine Sulfafurazole Sulfaguanidine Sulfamethazine Sulfadimethoxazole Sulfamethoxine Sulfamonomethoxine Sulfanilamide Sulfaquinoxaline *Sulfonamides and Diaminopyrimidines* Sulfamethoxypyridazine Trimethoprim+ Sulfonamide *Diaminopyrimidines* Baquiloprim Trimethoprim	*Quinolones**1G* Flumequin Miloxacin Nalidixic acid Oxolinic acid *Quinolones**2G (Fluoroquinolones)* Ciprofloxacin Danofloxacin Difloxacin Enrofloxacin Marbofloxacin Norfloxacin Ofloxacin Orbifloxacin	Enramycin Gramicidin Bacitracin *Polypeptides* *cyclic* Colistin Polymixin	*Cephalosporin 1G* Cefacetrile Cefalexin Cefalotin Cefapyrin Cefazolin Cefalonium *Cephalosporin 2G* Cefuroxime *Cephalosporin 3G* Cefoperazone Ceftiofur Ceftriaxone *Cephalosporin 4G* Cefquinome	Florphenicol Thiamphenicol	Pirlimycin Lincomycin
**Pleuromutilins** **48.6%**	**Ionophores** **42.9%**	**Novobiocin** **31.4%**	**Ansamycin-Rifamycins** **30%**	**Fosfomycin** **7.1%**	**Streptogramins** **5.7%**	**Quinoxalines** **4.3%**	**Orthosomycins** **4.3%**	**Fusidic Acid** **1.4%**	**Bicyclomycin** **1.4%**
Tiamulin Valnemulin	Lasalocid Maduramycin Monensin Narasin Salinomycin Semduramicin	Novobiocin	Rifampicin Rifaximin	Fosfomycin	Virginiamycin	Fusidic acid	Lasalocid Maduramycin Monensin Narasin Salinomycin Semduramicin	Fusidic acid	Bicozamycin

Adapted from OIE, List of antimicrobials of veterinary importance [27].

**Table 2 animals-12-00060-t002:** Characteristics of the reviewed studies describing the number of articles found by continent and country to which they belong.

Geographical Area	*n* = 165	Data Source Animal/Environment	*n* = 165	Data Source for Livestock	*n* = 165
North America ^(a)^	30	Livestock	112	Beef cattle	10
South America ^(b)^	33	Soil	34	Dairy cattle	29
Europe ^(c)^	31	Wastewater	19	Pork	23
Asia ^(d)^	35			Chicken	19
Africa ^(e)^	26			Egg	12
Oceania ^(f)^	10			Milk	32
				Sheep meat	15
				Fish	13
				Shrimp	12

(a) North America: Canada (*n* = 4), USA (*n* = 14), Mexico (*n* = 12). (b) South America: Peru (*n* = 6), Chile (*n* = 6), Venezuela (*n* = 8), Colombia (*n* = 9), Brazil (*n* = 1), Ecuador (*n* = 2), Argentina (*n* = 2). (c) Europe: Denmark (*n* = 3), Germany (*n* = 6), France (*n* = 3), the Netherlands (*n* = 2), Austria (*n* = 1), Spain (*n* = 7), UK (*n* = 3), Romania (*n* = 4 ), Italy (*n* = 1), Turkey (*n* = 1). (d) Asia: Vietnam (*n* = 5), China (*n* = 12), Israel (*n* = 1), Bangladesh (*n* = 4), Iraq (*n* = 5), Turkey (*n* = 2), Pakistan (*n* = 2), Singapore (*n* = 1), Iran (*n* = 3), India (*n* = 2). (e) Africa: Ghana (*n* = 2), Algeria (*n* = 1), Tanzania (*n* = 2), Egypt (*n* = 5), Sudan (*n* = 1), South Africa (*n*= 3), Nigeria (*n* = 5), Madagascar (*n* = 1), Ethiopia (*n* = 1), Morocco (*n* = 1), Tunisia (*n* = 2), Kenya (*n* = 2). (f) Oceania: Australia (*n* = 7), New Zealand (*n* = 3).

## Data Availability

Upon request to the corresponding author.

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
