# Peer review of "Worldwide Traceability of Antibiotic Residues from Livestock in Wastewater and Soil: A Systematic Review"

_animals, 2021, doi:10.3390/ani12010060_

Round 1
Reviewer 1 Report
I am quite confused with this manuscript. According to the submission report it is supposed to be a review but it is structured like a retrospective study with sections for introduction, materials and methods, results, discussion and conclusions. I am reviewing this as an original research article.
Here are my comments:
Line 28: “A database was created from studies specifying antibiotic residues worldwide and the papers used span the years 2000 to 2019. consisting of 173 papers, reporting the concentration of antibiotic residues found in the environment, livestock production and their products (milk and eggs) , wastewater and soil. ”. This sentence does not make sense. Please correct.
Line 35: “The main antibiotics used worldwide and in 35 animal production are sulfonamides, tetracyclines, quinolones, penicillin and cephalosporins”. Please add “according to our study” or something similar at the beginning of the sentence to specify that these is part of what you did in this project.
Line 36: “At present, although the use of antibiotics is being banned worldwide…”. This is simply not true. Antibiotics are used daily to treat diseases in animals and humans. Additionally, in food production, at least in the US , antibiotics are still in use. In cattle for example, animals are treated prophylactically with antibiotics to avoid bovine respiratory diseases. Please rephrase
Line 51: “The common antibiotics used in veterinary medicine [11,12] are excreted and disposed over the sewage system and in some cases to sewage treatment plants [13]”. Only the common antibiotics do so? What happens with the uncommon? I know what you mean but be specific with your writing. Please rephrase.
Line 55: “Antibiotic residues in food of animal origin also pose health risks, such as bacterial resistance, toxicity, hypersensitivity reactions, cancer, and teratogenicity [14]”. In lines 52-53 you talked about the “polluting effects on water and soil with negative effects on agricultural activities”, and here you only mention food animals. Do you think that the antibiotic residue in crops will also represent a health risk? I am sure there are studies about this. Please rephrase and add correct citations
Line 61: “Thus, member countries of the European Union, Mexico, New Zealand and the Republic of Korea banned the use of antibiotics as growth promoters”. Add when this happened and add corresponding citations. Other 60 countries such as Australia, Canada, Japan and the United States banned the use of antibiotics in animal production in 2014. Again, I do not think this completely true. Please take a look of this https://www.fda.gov/media/109457/download and rewrite accordingly
Line 64: “The greatest uncertainty about antibiotic use in livestock is found in low-income countries [15].” This sentence is empty. Uncertainty in terms of what? Because you don’t know what is going on in this countries or because they use to much? Please explain
Line 76: The information search focused on studies reporting veterinary antibiotic residues found in the environment, wastewater, soil…”. I believe, soil is part of the environment so please rephrase. You can use “residues found in environmental samples such as soil, water, plants…”
Line 89-97: This paragraph and table require a different header because they are not part of your “Search strategy and selection criteria”. I was actually thinking that they could fit better in the results sections.
Table 1: Please explain what the percentage s mean and why some of the antibiotics appear in bold.
Line 104: “…environment, livestock (cow, sheep, pig, horse, chicken, rabbit, goat), aquatic and terrestrial animal tissues, animal by-products (milk and eggs),wastewater and soil”. Again, what do you mean when you say environment? Because soil is part of the environment. Please rephrase
Line 113: “3.1. Veterinary antibiotics as pollutants in different continents” Please add a paragraph specifying how many manuscripts where use per continent and how many different countries were represented in this publications. That will help the reader to see if your results are representative for the entire continent (if you have only 10 manuscripts for Africa and they are all from congo, that would not be very representative, don’t you think?).
Line 114: “Bulleted lists look like this:” I do not understand why this is in here. Please remove
Figure 1: Some antibiotic names are missing in the legends. For example, in Asia, Africa and Europe Tetracycline is missing. Choose one color for each antibiotic class and use it consistently in all pie charts (e.i. bright green for tetracycline in all charts), and then use one common legend at the bottom of the figure. Please edit this figure accordingly and apply this changes to the other figures too.
ALL FIGURES. Figure presentation is very poor and not consistent. All pie charts are different size and in figure 3 the Eggs pie chart is a different style. Be consistent and use one seize and style across the manuscript. Same with the colors and the legends. For example, in figure 4 in the water pie sulfonamides appear in yellow while in the soil chart they appear in light grey. Again, be consistent because otherwise it gets really confusing and difficult to follow. Edit all figures accordingly.
Line 115: “In general, the main antibiotics used worldwide are Tetracyclines (25.5%), Sulfonamides (25.2%), Quinolones (13.5%) and Penicillins (8.5%).” How are this percentages obtained? And what do they mean? Please explain in detail in materials and methods. I am missing some statistical analysis to show if there are significant differences in antibiotic use between the different continents. Are there significant differences between different countries inside each continent?
Line 117: “Aminoglycosides among other antibiotics are less than the percentage of Penicillins when compared worldwide.” I do not understand this sentence. Where is the graph that shows results worldwide? Is the difference significant?
Line 128-131: Is this paragraph based on your analysis? Did you do the calculations? If not, this should go to discussion.
Line 132-135: How does this relate to your results? Please move it to your discussion
Line 136: “3.2. Residues of veterinary antibiotics in animal products and derivatives” As for the previous result section, please add a paragraph specifying how many manuscripts where use per animal product (you can also add a table). That will help the reader to see if your results are representative. Have you seen differences in antibiotic concentrations in animal products between the different continents? Please add that information to the text and maybe a figure.
Line 138: “In general, the antibiotics with the highest incidence of use are: Tetracyclines (22.2%), Sulphonamides (20.5%), Quinolones (16.2%), Penicillin (16.1%) and Aminoglycosides (10.7%).” Very poor result section you can have a much deeper analysis. Are there similarities between what found in cows and in milk? Same for egss and chicken? Same for fish and shrimp? Are there differences in the patterns observed for the different animal species? Or between terrestrial and aquatic animals? Are those differences significant?
Figure 3 does not help either. Re-organize the figure having products coming from same animal species close so you can compare them. I will remove the “Others” pie and just describe the results on the text.
Line 153: “3.3. Impact of antibiotic residues in water and soil”. Please Specify in the text that is wastewater. Where is the soil coming from? Farms? Same for the water, is it wastewater from animal producing facilities or from human environments? You need to explain all that in the text so the reader can connect what you are doing in here with the previous sections. As for the previous result sections, please add a paragraph specifying how many manuscripts where use per sample type (water and soil) (you can also add a table). Have you seen differences in antibiotic concentrations in soil and wastewater between the different continents? Please add that information to the text and maybe a figure.
Line 161: 4.1. Veterinary antibiotics as pollutants in different continents”In this section I was expecting to have a discussion of your results from figure 1 and results section 3.1. But you jumped to talk about figure 2 and some more facts not directly replated with the results of section 3.1. Please discuss section 3.1 appropriately in the text. Below some comments that may help you:
Line 162: “The largest use of veterinary antibiotics worldwide are China (45%), Brazil (7.9%), 162 USA (7.0%), Thailand (4.2%), India (2.2%), Iran (1.9%), Spain (1.9%), Russia (1.8%), Mexico 163 (1.7%) and Argentina (1.5%) [21].” Is this based on your results or previously published by citation 25? If it is part of your work remove citation. If it is part of previous research how does that correlate to your results? Is this similar to what you are seeing in your study?
Line 162-165: “The largest use of veterinary antibiotics worldwide are China (45%), Brazil (7.9%), 162 USA (7.0%), Thailand (4.2%), India (2.2%), Iran (1.9%), Spain (1.9%), Russia (1.8%), Mexico 163 (1.7%) and Argentina (1.5%) [21]. It is important to note that there is a particular distribu- 164 tion of antibiotic use by geographical area depending on their policies, economic/market 165 conditions, and dietary habits [25].” You have not looked at differences between countries so why bringing this as your first paragraph of the discussion? So, if you have differences between countries inside the same continent, do you think your data (that englobes all countries in continents) is a good representative? Please discuss. What about countries that have a protective policy for the use of antibiotics? Do they show same antimicrobial residues than the ones who does not? Add a result section about this if required and discuss
Line 167-173: “In India, in the last decade, changes in the population diet generated by the improved 167 standards of living, has led to a demand for animal protein with consequent intensifica- 168 tion of pig, poultry and fish farming [26], which has had a significant effect on the use of 169 antibiotics such tetracyclines, penicillin’s, sulfonamides“. In your analysis you do not show results for India. You show results for Asia. Is India a good representative of what happens in Asia? Please discuss. Are those measurements similar in China, Russia, Iran and Thailand as high antimicrobial consumers? Are those counties under some type of policy against the excessive use of antibiotics? Is it the same in countries from Asia that have a protective policy for the use of antibiotics for veterinary medicine? Please discuss in the text.
Line 175: “In Africa, the main antibiotics used for livestock farming are tetracyclines, fluoro- 175 quinolones and β-lactams / aminoglycosides (33.6%, 26.5% and 20.4%, respectively) [29]. You have looked at this in your study. Why don’t you mention what you found? Is it similar to what observed in citation 29? Please discuss in the text. How does this compare to Asia on the other contients? Do they have preotective messurements for the use of antibiotics for veterinary medicine? Please discuss
Line 177: “Factors influencing the 177 excessive use of antibiotics in Africa are the accessibility to purchase drugs, as well as the 178 types of animals kept (e.g. poultry) and the farming system of the animals (e.g. intensive); 179 however, farmers had limited experience on their use for animal production [6].” Your results are based in percentages and do not compare Africa to other continents so, how do you know based on your analysis that Africa uses antibiotics excessively? Add a context to this discussion point based on your data.
Line 184: “the main recipients found in this part of the world.” What do you mean with main recipients?
Line 184: “Although the use of antibiotics in the 184 EU is more restricted and monitored, some member countries still use large quantities, for 185 example, countries such as Spain and Italy consume significantly more drugs than North- 186 ern members of the EU [31]. “ How does this correlate to your results? Do you see differences between Europe and other continents?
Line 188: ”When talking about meat producers, one must mention the world's leading meat pro- 188 ducers, North and South America, which are also among the main consumers of veteri- 189 nary antibiotics [32,33]. For example, the United States uses 24.6 million pounds of anti- 190 microbials for non-therapeutic purposes in chickens, cattle, and pigs, the most commonly 191 used being tetracyclines, penicillin’s and sulfonamides [32] and Brazil, being the fourth 192 largest pork producer in the world, uses sulfonamides [33]. However, although govern- 193 ment entities are trying to regularize the use of antibiotics, producers are not willing to 194 stop using them, as they mention that it would be impossible to sustain current market 195 demands without large quantities of antimicrobials [33]. Please put this in the context of your results. Are the patterns observed in North and South America more similar among each other than when compared to the other continents? Please discuss
Line 197: “Although most countries have joined in the non-misuse of antibiotics in food animals, 197 publication of reports, complex political, economic, and social barriers still limit the qual- 198 ity of data on this issue [31,32]. Data on their use of antibiotics is readily available from 199 countries that export a significant part of their animal production than those countries 200 where the majority their production is destined for domestic market [24]. ” How did this affect your data analysis in this project? Could this be one of your limitaions? Please discuss.
Line 212-221: “Overuse or lack of control in antibiotic stewardship results in high antibiotic deposi- 212 tion in the animal and excretion into the environment [37]. Olatoye and Ehinmoro [38] 213 detected approximately 54 % of oxytetracycline residues in livestock. In Egypt, in fresh 214 chicken meat and liver samples, 44% of the samples contained tetracycline residues, rang- 215 ing from 38% to 52%, and the corresponding contamination residues ranged from 103 216 μg/kg to 8148 μg/kg, which are above the Codex maximum residue limit (200 μg/kg and 217 600 μg/kg for chicken meat and liver, respectively, expressed as the sum of the tetracycline 218 group) [39]. The European Union (EU) No 37/2010 has stipulated that the maximum resi- 219 due limit for all synthetic antimicrobials such as sulphonamides is 100 μg/kg in edible 220 animal tissue [40]. According to FDA reports, tetracyclines show the highest level of drug 221 application, followed by sulfonamides and aminoglycosides [41].” How does this all correlate to your work? You do not present any antimicrobial concentration value in your analysis, and you do not show differences in residues found in animal tissues between countries? Add this analysis to your result section or remove this paragraph from the discussion.
Line 223: “It has been mentioned that pigs are the main consumers of antimicrobials and are 223 expected to use 45% of antimicrobials for animal production from 2017 to 2030, with cattle 224 using 22% and chickens 33% of antimicrobials. On average, pigs consume 193 mg / PCU 225 (stock correction unit), cattle consume 42 mg / PCU of antimicrobials and chickens con- 226 sume 68 mg / PCU of antimicrobials [24]. Same as comment above, how does this all correlate to your work? Explain or remove from discussion
Line 235: “This may be because in veterinary practice, sulfonamides are widely 235 used due to their broad spectrum of activity and low cost as well as to promote the growth 236 of animals [43,44]. “ Are they use widely in veterinary practice to treat cattle or all animal species in food production? Cattle is not the only animal species with high levels of sufonamides, pigs, chicken and shrimp too. What happens with them?
Line 244: “Pereira et al. [46] evaluated the effect of antimicrobial use on drug resistance in fecal 244 E. coli isolated from pre-weaned dairy calves and found that isolates from calves treated 245 with enrofloxacin were more likely to be resistant to fluoroquinolones. This example in- 246 dicates an important concern, as these antimicrobial agents are essential in human medi- 247 cine for treatment against Salmonella, Campylobacter and Shigella species. There are also 248 other reports of antibiotic and anthelmintic residues in dairy products [47,48]. “How does this all correlate to your work? Explain or remove from discussion
Line 250: “The use of antibiotics as growth promoters in broilers is intended to modify the in- 250 testinal flora, thereby improving feed absorption to increase muscle mass by 15-20% in a 251 short time. In addition, antibiotics are commonly used in high concentrations in over- 252 crowded poultry [49].” How does this all correlate to your work? Explain or remove from discussion
Line 254: ” The selection pressure for resistant bacteria in poultry is high, and consequently the 254 content of their fecal flora has a relatively high proportion of resistant bacteria. Van den 255 Bogaard et al. [50] analyzed antimicrobial resistance in feces from turkeys, broilers and 256 chickens producing eggs for human consumption, where they found strong evidence of 257 the spread of antibiotic-resistant E. coli. Resistant faecal E. coli from poultry can infect 258 humans directly, via farmers and food, for example when eggs are contaminated during 259 laying.” Your study do not look at bacterial populations resistant to antibiotics, and although I find that topic fascinating, I do not think it is suitable to discus in depth in here.
Line 262:” The marketing of antibiotics for veterinary use is often not adequately restricted, as 262 there are cases of empirical supply without prescription. In addition, there is a lack of 263 adequate registration in the control of medicines by the health authorities as well as peri- 264 odic monitoring of infectious agents with zoonotic potential in backyard, semi-intensive 265 and intensive production sites. Good production practices are not considered by all pro- 266 ducers from prevention, treatment and slaughter, due to lack of maintenance of facilities 267 [52].” How does this all correlate to your work? Explain or remove from discussion
Line 271: “Oxytetracy- 270 cline, florfenicol, sarafloxacin, erythromycin and sulfonamides are widely used in aqua- 271 culture and are therefore detected in aquatic samples [54]. “ is this what you saw in your analysis? Discuss your results!
Line 283: “4.3. Impact of antibiotic residues in water and soil “ Please specify is waste water.
Line 288: “In the United States (US), 70% of antibiotics were used in animal agriculture, which 288 is eight times the amount used in human medicine [58]; in 2013, US livestock producers 289 purchased 14,900 tons of antimicrobials [59]. Van Boeckel et al. [19] estimate that from 290 2010 to 2030, antibiotic use in food animal production will increase by 67% from 63,151 ± 291 1,560 tons to 105,596 ± 3,605 tons. “ how does this all correlate to your work? Explain or remove from discussion
Line 311: “It has been shown through molecular markers that the main cause of antibiotic resi- 311 dues in the environment is due to livestock feces, as doses much higher than prescribed 312 are applied to the human population, as demonstrated in animal excrement in the United 313 States, where values exceeded 100 times more than those found in wastewater of anthro- 314 pogenic origin [70]. “ Does this match your results? Are the same antibiotics in water than in animals? Please Discuss
Line 316: “It is important to control the doses used in livestock production to avoid pathogen 316 resistance to antibiotics and water contamination, as wastewater in some areas is used as 317 an alternative for irrigation of agricultural fields. “ Add reference
Line 318: “According to a study by Xu et. al [71] in 318 urban rivers in Beijing, China, the sources of contamination are in places where large 319 amounts of antibiotics are used, such as hospitals.” You are looking at livestock not humans, How does this all correlate to your work? Explain or remove from discussion
Line 332: “Antibiotics can affect bacterial enzyme activity including dehydrogenases, phospha- 332 tases and ureases which are considered as indicators of soil biological activity. With the 333 presence of antibiotics in the environment this may favor the increase of pathogens or 334 parasites in soil and water. For example, the presence of antibiotic contaminants in water 335 favors the increase of toxic Cyanobacteria species that cause the phenomenon of eutroph- 336 ication in freshwater systems and with it possible risks to human health [74]. “Your study do not look at bacterial populations resistant to antibiotics, and although I find that topic fascinating, I do not think it is suitable to discus in depth in here.
Line 338: “With the development and improvement of laboratory analytical techniques, 338 changes to international and national legislation have been promoted. For example, the 339 European Commission [20,75], US Food and Drug Administration [76], and Codex Ali- 340 mentarius [77] have established maximum residue limits (MRLs) for VAs in foodstuffs 341 from animal origin [78]. Therefore, in recent years, rapid methods with the advantage of 342 easy performance, high sensitivity and high throughput are being proposed and used ex- 343 tensively. 344 “How does this all correlate to your work? Explain or remove from discussion
Line 347: “The use of antibiotics is defined 347 by the different local policies and legislation in each region according to the residual ef- 348 fects they have on the environment and their possible effects on human health” How can you conclude this if you have not looked at the differences between countries or cmunities?
Line 349: “At present, 349 although the use of antibiotics is being banned worldwide, producers still do not want to 350 stop using them as they mention that it would be impossible to sustain the current market 351 demands without large quantities of antimicrobial. : This is exactly what you said in your introduction, so you knew that before doing your study and is also not true. This is not a discussion. What about the residue in animal tissues and in environmental samples? nothing to conclude about that?
Author Response
Dear editor. and reviewers, we are very grateful for your comments, which have substantially improved this work. The. study. has. been. reanalysed. according. to. the. suggestions. of. reviewer. 1,. as. we. had. the. data. but. did. not. know. how. to. present. them. This. new. version. has. taken. more. time. as. we. had. to. reanalyse. and. present. the. data. and. include. the. list. of. articles. (table. 1S). and. the. number. of. studies. by. continent,. environment. and. animal. species. which. ultimately. eliminated. the. few. data. we. had. for. equines. and. rabbits.
We hope that the reviewers' comments have been addressed and look forward to hearing from you if you have any comments.
The authors
Reviewers 1
I am quite confused with this manuscript. According to the submission report it is supposed to be a review but it is structured like a retrospective study with sections for introduction, materials and methods, results, discussion and conclusions. I am reviewing this as an original research article.
Here are my comments:
Q Line 28: “A database was created from studies specifying antibiotic residues worldwide and the papers used span the years 2000 to 2019. consisting of 173 papers, reporting the concentration of antibiotic residues found in the environment, livestock production and their products (milk and eggs), wastewater and soil.
This sentence does not make sense. Please correct.
R I has been re write it, Our database was composed of 173 articles, reporting the concentration of antibiotic residues found in the environment, livestock (cow, sheep, pig, horse, chicken, rabbit, goat), aquatic and terrestrial animal tissues, animal products (milk and eggs), wastewater and soil. The documents were obtained from Asia, Africa, North America, South America, Europe and Oceania. A descriptive analysis of antibiotic residues found worldwide was analyzed according to each of the variables used such as antibiotic family, name, concentration (% and mg/kg or ppm), and country and continent where the residue was found. The descriptive analysis was carried out using the “describe” function of psych package and pirate plots were drawn.
Line 35: “The main antibiotics used worldwide and in animal production are sulfonamides, tetracyclines, quinolones, penicillin and cephalosporins”.
Please add “according to our study” or something similar at the beginning of the sentence to specify that these is part of what you did in this project.
R has been include it
Line 36: “At present, although the use of antibiotics is being banned worldwide…”. This is simply not true. Antibiotics are used daily to treat diseases in animals and humans. Additionally, in food production, at least in the US , antibiotics are still in use. In cattle for example, animals are treated prophylactically with antibiotics to avoid bovine respiratory diseases. Please rephrase
R you. are right. sentence. has been change it,
Antibiotic residues in food of animal origin also pose health risks, such as bacterial resistance, toxicity, hypersensitivity reactions, cancer, and teratogenicity [15]. In May 2015, the 68th World Health Assembly recognized the importance of the antimicrobial resistance and adopted a plan to reduce the unnecessary use of antimicrobials in humans and animals [16,17]. Thus, since 2006, member countries of the European Union [18], and since 2011, New Zealand and the Republic of Korea banned the use of antibiotics as growth promoters [19]. Other countries such as Australia, Canada, Japan and the United States have applied policies and regulations to ensure that they are only used by licensed veterinarians [20-22].
Line 51: “The common antibiotics used in veterinary medicine [11,12] are excreted and disposed over the sewage system and in some cases to sewage treatment plants [13]”. Only the common antibiotics do so? What happens with the uncommon? I know what you mean but be specific with your writing. Please rephrase.
R has. been re write it , All antibiotics used in veterinary medicine [11,12] are excreted and disposed over the sewage system and in some cases to sewage treatment plants [13].
Line 55: “Antibiotic residues in food of animal origin also pose health risks, such as bacterial resistance, toxicity, hypersensitivity reactions, cancer, and teratogenicity [14]”. In lines 52-53 you talked about the “polluting effects on water and soil with negative effects on agricultural activities”, and here you only mention food animals. Do you think that the antibiotic residue in crops will also represent a health risk? I am sure there are studies about this. Please rephrase and add correct citations
R has been re write it, This situation has polluting effects on water and soil with negative effects on agricultural activities, for example, it has been mentioned that in the soil, antibiotic residues are responsible for anoxic denitrification since they affect bacterial communities responsible for this process [14].
Line 61: “Thus, member countries of the European Union, Mexico, New Zealand and the Republic of Korea banned the use of antibiotics as growth promoters”.
Add when this happened and add corresponding citations.
Other 60 countries such as Australia, Canada, Japan and the United States banned the use of antibiotics in animal production in 2014.
Again, I do not think this completely true. Please look of this
R has. been. change it.
, In May 2015, the 68th World Health Assembly recognized the importance of the antimicrobial resistance and adopted a plan to reduce the unnecessary use of antimicrobials in humans and animals [16,17]. Thus, since 2006, member countries of the European Union [18], and since 2011, New Zealand and the Republic of Korea banned the use of antibiotics as growth promoters [19]. Other countries such as Australia, Canada, Japan and the United States have applied policies and regulations to ensure that they are only used by licensed veterinarians [20-22]. Large meat producing countries such as Argentina, Brazil, China, India, Indonesia, the Philippines, Russia and South Africa have not banned the use of antibiotics as growth promoters [23]. The greatest uncertainty about antibiotic use in livestock is found in low-income countries, due to the lack of information on the use of antibiotics [16].
Line 64: “The greatest uncertainty about antibiotic use in livestock is found in low-income countries [15].” This sentence is empty. Uncertainty in terms of what? Because you don’t know what is going on in these countries or because they use to much? Please explain
R The greatest uncertainty about antibiotic use in livestock is found in low-income countries, due to the lack of information on the use of antibiotics [16].
Line 76: The information search focused on studies reporting veterinary antibiotic residues found in the environment, wastewater, soil…”. I believe, soil is part of the environment so please rephrase. You can use “residues found in environmental samples such as soil, water, plants…”
R Therefore, this review focuses on the trend of research work on antibiotic residues found in environmental samples such as water, soil and livestock products, aquatic and terrestrial animal tissues, and animal products (milk and eggs).
Line 89-97: This paragraph and table require a different header because they are not part of your “Search strategy and selection criteria”.
I was actually thinking that they could fit better in the results sections.
R The search strategy. we leave in M&M; describing how we looking for the information, later this part is discussed.
Table 1: Please explain what the percentages mean and why some of the antibiotics appear in bold.
R, The ranking order of antibiotic families based on their occurrence (%) is shown in Table 1 [27], antibiotics highlighted in bold letters represent their family and the most commonly used antibiotics, respectively.
Line 104: “…environment, livestock (cow, sheep, pig, horse, chicken, rabbit, goat), aquatic and terrestrial animal tissues, animal by-products (milk and eggs),wastewater and soil”. Again, what do you mean when you say environment? Because soil is part of the environment. Please rephrase
R has been re write as: The keywords used were antibiotic residues in the environment (wastewater and soil), traceability, animal husbandry, animal species (cow, sheep, pig, horse, chicken, rabbit, goat),
Line 113: “3.1. Veterinary antibiotics as pollutants in different continents” Please add a paragraph specifying how many manuscripts where use per continent and how many different countries were represented in this publications. That will help the reader to see if your results are representative for the entire continent (if you have only 10 manuscripts for Africa and they are all from congo, that would not be very representative, don’t you think?).
R Very. good. observation, we have performed a table 2 , showing the Characteristics of the reviewed studies describes the number of articles found by continent and country to which they belong.
Line 114: “Bulleted lists look like this:” I do not understand why this is in here. Please remove
R has been removed
Figure 1: Some antibiotic names are missing in the legends. For example, in Asia, Africa and Europe Tetracycline is missing. Choose one color for each antibiotic class and use it consistently in all pie charts (e.i. bright green for tetracycline in all charts), and then use one common legend at the bottom of the figure. Please edit this figure accordingly and apply this changes to the other figures too.
ALL FIGURES. Figure presentation is very poor and not consistent. All pie charts are different size and in figure 3 the Eggs pie chart is a different style. Be consistent and use one seize and style across the manuscript. Same with the colors and the legends. For example, in figure 4 in the water pie sulfonamides appear in yellow while in the soil chart they appear in light grey. Again, be consistent because otherwise it gets really confusing and difficult to follow. Edit all figures accordingly.
R all figures has been reedited, after. your comment we reanalyze it, the using the ppm concentration, this data we had it, but we didn’t know how to analyze it and show it, now we believe that the presentation of the results is better and more confident.
Line 115: “In general, the main antibiotics used worldwide are Tetracyclines (25.5%), Sulfonamides (25.2%), Quinolones (13.5%) and Penicillins (8.5%).” How are this percentages obtained? And what do they mean? Please explain in detail in materials and methods. I am missing some statistical analysis to show if there are significant differences in antibiotic use between the different continents. Are there significant differences between different countries inside each continent
R Your question is very good, Originally we could only present results by. frequency of. occurrence of the results, then after your. comments, we decided to re-analyse the results based on the. ppm. concentrations of the antibiotics found in the studies, and from there we proceeded to analyse the results again. Then… The descriptive analysis was carried out using the “describe” function of psych package [28] and pirate plots were drawn using the Yarrr package [29]
Now we cannot analyze if there are significant differences because we are not dealing with studies that have a control (without antibiotics). vs a treatment (with antibiotics) because we tried to look for studies that showed the presence of antibiotics where there should not be.
Line 117: “Aminoglycosides among other antibiotics are less than the percentage of Penicillins when compared worldwide.” I do not understand this sentence. Where is the graph that shows results worldwide? Is the difference significant?
R Figure 2 has been performed, indicating the concentration in ppm between the different continents, instead of percentages, and the redaction of the results has been change it
Line 128-131: Is this paragraph based on your analysis? Did you do the calculations? If not, this should go to discussion.
R Redaction has been change it, , we. made the calculations, the data showed in the figures, has been analyzed and performed by out team
Line 132-135: How does this relate to your results? Please move it to your discussion
R has been rewrited, the results showed in the figures, has been preformed by ourselves
Line 136: “3.2. Residues of veterinary antibiotics in animal products and derivatives” As for the previous result section, please add a paragraph specifying how many manuscripts where use per animal product (you can also add a table). That will help the reader to see if your results are representative. Have you seen differences in antibiotic concentrations in animal products between the different continents? Please add that information to the text and maybe a figure.
R Very good observation, this table has been preforme as a supplementary table 1S, and has been include it the whole manuscripts used in the search data base.
Line 138: “In general, the antibiotics with the highest incidence of use are: Tetracyclines (22.2%), Sulphonamides (20.5%), Quinolones (16.2%), Penicillin (16.1%) and Aminoglycosides (10.7%).” Very poor result section you can have a much deeper analysis. Are there similarities between what found in cows and in milk? Same for egss and chicken? Same for fish and shrimp? Are there differences in the patterns observed for the different animal species? Or between terrestrial and aquatic animals? Are those differences significant?
R Has been rewrite In in deep, Unfortunately we cannot make a correlation analysis to indicate whether or not there is a. relationship between dairy cow and milk or. hens and. eggs, because they come from. different studies, on the one hand some author. determines the concentration of certain antibiotics in milk. or eggs. and. in. another study, perhaps in another continent another author determines the. concentration of other antibiotics in. poultry, fish or. dairy cattle.
or at least. our. group. would. not. know. how. to. analyze. it,. as. there. must. be. a. minimum. number. of. studies. presenting. control. vs. treatment. The number of animals and their SD. of each study... only then we could think of a meta-analysis.
Figure 3 does not help either. Re-organize the figure having products coming from same animal species close so you can compare them. I will remove the “Others” pie and just describe the results on the text.
R has ben. reorganized. and improved this secction
Line 153: “3.3. Impact of antibiotic residues in water and soil”. Please Specify in the text that is wastewater. Where is the soil coming from? Farms? Same for the water, is it wastewater from animal producing facilities or from human environments? You need to explain all that in the text so the reader can connect what you are doing in here with the previous sections. As for the previous result sections, please add a paragraph specifying how many manuscripts where use per sample type (water and soil) (you can also add a table). Have you seen differences in antibiotic concentrations in soil and wastewater between the different continents? Please add that information to the text and maybe a figure.
R This section. has been. rewrite it according. to. you suggestion
Line 161: 4.1. Veterinary antibiotics as pollutants in different continents”I n this section I was expecting to have a discussion of your results from figure 1 and results section 3.1. But you jumped to talk about figure 2 and some more facts not directly replated with the results of section 3.1. Please discuss section 3.1 appropriately in the text. Below some comments that may help you:
R Has been discussed , thank you for your suggestion
Line 162: “The largest use of veterinary antibiotics worldwide are China (45%), Brazil (7.9%), 162 USA (7.0%), Thailand (4.2%), India (2.2%), Iran (1.9%), Spain (1.9%), Russia (1.8%), Mexico 163 (1.7%) and Argentina (1.5%) [21].” Is this based on your results or previously published by citation 25? If it is part of your work remove citation. If it is part of previous research how does that correlate to your results? Is this similar to what you are seeing in your study?
R Has been rewrite it
Line 162-165: “The largest use of veterinary antibiotics worldwide are China (45%), Brazil (7.9%), 162 USA (7.0%), Thailand (4.2%), India (2.2%), Iran (1.9%), Spain (1.9%), Russia (1.8%), Mexico 163 (1.7%) and Argentina (1.5%) [21]. It is important to note that there is a particular distribu- 164 tion of antibiotic use by geographical area depending on their policies, economic/market 165 conditions, and dietary habits [25].” You have not looked at differences between countries so why bringing this as your first paragraph of the discussion? So, if you have differences between countries inside the same continent, do you think your data (that englobes all countries in continents) is a good representative? Please discuss. What about countries that have a protective policy for the use of antibiotics? Do they show same antimicrobial residues than the ones who does not? Add a result section about this if required and discuss
R this section has been rewrite it
The regulation of antibiotics in animal feed (for growth promotion or therapeutic use) is a priority for the Asian region, and policies for their prohibition have been developed, but few countries have the capacity to guarantee their application [36]. The Chinese government has implemented policies to control the use of antibiotics: the use of any antibiotics included in the prohibited list is banned, the use of antibiotics during the waiting period is prohibited, the purchase of antibiotics without veterinary prescription is prohibited, the use of medically im-portant antibiotics in food animals is prohibited, however they are still purchased without prescription and not monitored in most farms [37]. In Russia, farmers can use antibiotics with-out any restrictions, while some feed antibiotics are subject to state control [38]. Iran is one of the countries where the sale of antibiotics is not controlled, and no prescription is needed to purchase the drugs. This is due to the lack of national action plans to try to control this problem, low awareness of farm producers, fragmented information systems due to political problems, so that monitoring, and surveillance is irregular [39]. In Thailand, surveillance of antibiotic use is insufficient due to gaps in human resources, particularly for smallholder farmers, and there is little information from farmers on the resistance that can be caused by antibiotics misuse [40].
In India, in the last decade, changes in the population diet generated by the improved standards of living, has led to a demand for animal protein with consequent intensification of pig, poultry and fish farming [41], which has had a significant effect on the use of antibiotics such tetracyclines, penicillins, and sulfonamides. However, the highest concentrations of anti-biotics were found for cephalosporins and fluroquinolones (Fig. 2). This higher concentration may be related to the fact that these antibiotics are frequently used in swine production for respiratory problems as well as in aquaculture, and Asia allocates most of the veterinary anti-biotics to swine production [42]. This is mainly due to the fact that the use of antibiotics in the livestock sector in Asia presents the weakness or the lack of regulations, adequate policies and the implementation of quality standards, causing the emergence and spread of antibiotic re-sistance today [43,44].
In Africa, the main antibiotics used for livestock farming are tetracyclines, fluoroquin-olones and β-lactams/aminoglycosides (33.6%, 26.5% and 20.4%, respectively) [45]. In our study, the above mentioned can be confirmed since tetracyclines and fluoroquinolones are the most frequently found antibiotics in animal products (Fig. 2). Factors influencing the excessive use of antibiotics in Africa are the accessibility to purchase drugs, as well as the types of ani-mals kept (e.g. poultry) and the farming system of the animals (e.g. intensive); however, farmers had limited experience on their use for animal production [6]. Weak national surveillance sys-tems, lack of coordination between national authorities and the private sector, lack human re-sources, and insufficient regulatory standards are other reasons for the increase in the overuse of antibiotics in Africa [46].
Line 167-173: “In India, in the last decade, changes in the population diet generated by the improved 167 standards of living, has led to a demand for animal protein with consequent intensifica- 168 tion of pig, poultry and fish farming [26], which has had a significant effect on the use of 169 antibiotics such tetracyclines, penicillin’s, sulfonamides“. In your analysis you do not show results for India. You show results for Asia. Is India a good representative of what happens in Asia? Please discuss. Are those measurements similar in China, Russia, Iran and Thailand as high antimicrobial consumers? Are those counties under some type of policy against the excessive use of antibiotics? Is it the same in countries from Asia that have a protective policy for the use of antibiotics for veterinary medicine? Please discuss in the text.
R This section has been rewrite it
In India, in the last decade, changes in the population diet generated by the improved standards of living, has led to a demand for animal protein with consequent intensification of pig, poultry and fish farming [41], which has had a significant effect on the use of antibiotics such tetracyclines, penicillins, and sulfonamides. However, the highest concentrations of anti-biotics were found for cephalosporins and fluroquinolones (Fig. 2). This higher concentration may be related to the fact that these antibiotics are frequently used in swine production for respiratory problems as well as in aquaculture, and Asia allocates most of the veterinary anti-biotics to swine production [42]. This is mainly due to the fact that the use of antibiotics in the livestock sector in Asia presents the weakness or the lack of regulations, adequate policies and the implementation of quality standards, causing the emergence and spread of antibiotic re-sistance today [43,44].
Line 175: “In Africa, the main antibiotics used for livestock farming are tetracyclines, fluoro- 175 quinolones and β-lactams / aminoglycosides (33.6%, 26.5% and 20.4%, respectively) [29]. You have looked at this in your study. Why don’t you mention what you found? Is it similar to what observed in citation 29? Please discuss in the text. How does this compare to Asia on the other contients? Do they have preotective messurements for the use of antibiotics for veterinary medicine? Please discuss
R this section has been rewrite it, In Africa, the main antibiotics used for livestock farming are tetracyclines, fluoroquin-olones and β-lactams/aminoglycosides (33.6%, 26.5% and 20.4%, respectively) [45]. In our study, the above mentioned can be confirmed since tetracyclines and fluoroquinolones are the most frequently found antibiotics in animal products (Fig. 2). Factors influencing the excessive use of antibiotics in Africa are the accessibility to purchase drugs, as well as the types of ani-mals kept (e.g. poultry) and the farming system of the animals (e.g. intensive); however, farmers had limited experience on their use for animal production [6]. Weak national surveillance sys-tems, lack of coordination between national authorities and the private sector, lack human re-sources, and insufficient regulatory standards are other reasons for the increase in the overuse of antibiotics in Africa [46].
Line 177: “Factors influencing the 177 excessive use of antibiotics in Africa are the accessibility to purchase drugs, as well as the 178 types of animals kept (e.g. poultry) and the farming system of the animals (e.g. intensive); 179 however, farmers had limited experience on their use for animal production [6].” Your results are based in percentages and do not compare Africa to other continents so, how do you know based on your analysis that Africa uses antibiotics excessively? Add a context to this discussion point based on your data.
R yes. it has been rewrite it
Line 184: “the main recipients found in this part of the world.” What do you mean with main recipients?
R it has been rewrite it
Line 184: “Although the use of antibiotics in the 184 EU is more restricted and monitored, some member countries still use large quantities, for 185 example, countries such as Spain and Italy consume significantly more drugs than North- 186 ern members of the EU [31]. “ How does this correlate to your results? Do you see differences between Europe and other continents?
R Numerically, yes. There are differences between continents. as I mentioned we cannot make a statistical analysis between them, because. there is a great variety of studies. and each one. evaluates certain factors, however it can be observed that there is a difference in the use of antibiotics in each continent. this section has been rewritten and. discussed.
Line 188: ”When talking about meat producers, one must mention the world's leading meat pro- 188 ducers, North and South America, which are also among the main consumers of veteri- 189 nary antibiotics [32,33]. For example, the United States uses 24.6 million pounds of anti- 190 microbials for non-therapeutic purposes in chickens, cattle, and pigs, the most commonly 191 used being tetracyclines, penicillin’s and sulfonamides [32] and Brazil, being the fourth 192 largest pork producer in the world, uses sulfonamides [33]. However, although govern- 193 ment entities are trying to regularize the use of antibiotics, producers are not willing to 194 stop using them, as they mention that it would be impossible to sustain current market 195 demands without large quantities of antimicrobials [33]. Please put this in the context of your results. Are the patterns observed in North and South America more similar among each other than when compared to the other continents? Please discuss
R this section has been discussed
Line 197: “Although most countries have joined in the non-misuse of antibiotics in food animals, 197 publication of reports, complex political, economic, and social barriers still limit the qual- 198 ity of data on this issue [31,32]. Data on their use of antibiotics is readily available from 199 countries that export a significant part of their animal production than those countries 200 where the majority their production is destined for domestic market [24]. ” How did this affect your data analysis in this project? Could this be one of your limitaions? Please discuss.
R . Data on their use of antibiotics is readily available from countries that export a significant portion of their animal production than those countries where the majority their production is destined for domestic market [52]. All of the above, were limitations for obtaining more articles needed to reinforce the obtained information, and until now, consumption of antibiotics at present is still in discriminated.
Line 212-221: “Overuse or lack of control in antibiotic stewardship results in high antibiotic deposi- 212 tion in the animal and excretion into the environment [37]. Olatoye and Ehinmoro [38] 213 detected approximately 54 % of oxytetracycline residues in livestock. In Egypt, in fresh 214 chicken meat and liver samples, 44% of the samples contained tetracycline residues, rang- 215 ing from 38% to 52%, and the corresponding contamination residues ranged from 103 216 μg/kg to 8148 μg/kg, which are above the Codex maximum residue limit (200 μg/kg and 217 600 μg/kg for chicken meat and liver, respectively, expressed as the sum of the tetracycline 218 group) [39]. The European Union (EU) No 37/2010 has stipulated that the maximum resi- 219 due limit for all synthetic antimicrobials such as sulphonamides is 100 μg/kg in edible 220 animal tissue [40]. According to FDA reports, tetracyclines show the highest level of drug 221 application, followed by sulfonamides and aminoglycosides [41].” How does this all correlate to your work? You do not present any antimicrobial concentration value in your analysis, and you do not show differences in residues found in animal tissues between countries? Add this analysis to your result section or remove this paragraph from the discussion.
R Now. we have includle antibiotic concentration and the discussion. has ben improved
Line 223: “It has been mentioned that pigs are the main consumers of antimicrobials and are 223 expected to use 45% of antimicrobials for animal production from 2017 to 2030, with cattle 224 using 22% and chickens 33% of antimicrobials. On average, pigs consume 193 mg / PCU 225 (stock correction unit), cattle consume 42 mg / PCU of antimicrobials and chickens con- 226 sume 68 mg / PCU of antimicrobials [24]. Same as comment above, how does this all correlate to your work? Explain or remove from discussion
R has ben discussed
Line 235: “This may be because in veterinary practice, sulfonamides are widely 235 used due to their broad spectrum of activity and low cost as well as to promote the growth 236 of animals [43,44]. “ Are they use widely in veterinary practice to treat cattle or all animal species in food production? Cattle is not the only animal species with high levels of sufonamides, pigs, chicken and shrimp too. What happens with them?
R has been discussed
Line 244: “Pereira et al. [46] evaluated the effect of antimicrobial use on drug resistance in fecal 244 E. coli isolated from pre-weaned dairy calves and found that isolates from calves treated 245 with enrofloxacin were more likely to be resistant to fluoroquinolones. This example in- 246 dicates an important concern, as these antimicrobial agents are essential in human medi- 247 cine for treatment against Salmonella, Campylobacter and Shigella species. There are also 248 other reports of antibiotic and anthelmintic residues in dairy products [47,48]. “How does this all correlate to your work? Explain or remove from discussion
R Pereira et al. [61] evaluated the effect of antimicrobial use on drug resistance in fecal E. coli isolated from pre-weaned dairy calves and found that isolates from calves treated with enrofloxacin were more likely to be resistant to fluoroquinolones. This example indicates an important concern, as these antimicrobial agents are essential in human medicine for treatment against Salmonella spp., Campylobacter spp. and Shigella spp. There are also other reports of an-tibiotic and anthelmintic residues in dairy products [69,70] and tetracyclines are one of the main antibiotics found in dairy products according to the data collected in this article.
Line 250: “The use of antibiotics as growth promoters in broilers is intended to modify the in- 250 testinal flora, thereby improving feed absorption to increase muscle mass by 15-20% in a 251 short time. In addition, antibiotics are commonly used in high concentrations in over- 252 crowded poultry [49].” How does this all correlate to your work? Explain or remove from discussion
R The use of antibiotics as growth promoters in broilers is intended to modify the intestinal flora, thereby improving feed absorption to increase muscle mass by 15-20% in a short time. In addition, antibiotics are commonly used in high concentrations in overcrowded poultry [71]. This can be seen in our work, as the most common antibiotic found in chickens were fluoroquin-olone, tetracycline, and aminoglycoside. It is important to note that although many antibiotics are used more frequently in animal production, the residues found in products intended for human consumption of animal origin were different from those found in the present work, this may be influenced because the persistence of antibiotic residues is affected by the animal spe-cies from which they come, their respective diet and intestinal microbiota, in addition to the fact that some antibiotics degrade more rapidly (hours) [72] with respect to others, the time that the antibiotic was not administered until the animals were slaughtered [73].
Line 254: ” The selection pressure for resistant bacteria in poultry is high, and consequently the 254 content of their fecal flora has a relatively high proportion of resistant bacteria. Van den 255 Bogaard et al. [50] analyzed antimicrobial resistance in feces from turkeys, broilers and 256 chickens producing eggs for human consumption, where they found strong evidence of 257 the spread of antibiotic-resistant E. coli. Resistant faecal E. coli from poultry can infect 258 humans directly, via farmers and food, for example when eggs are contaminated during 259 laying.” Your study do not look at bacterial populations resistant to antibiotics, and although I find that topic fascinating, I do not think it is suitable to discus in depth in here.
R has been deleted
Line 262:” The marketing of antibiotics for veterinary use is often not adequately restricted, as 262 there are cases of empirical supply without prescription. In addition, there is a lack of 263 adequate registration in the control of medicines by the health authorities as well as peri- 264 odic monitoring of infectious agents with zoonotic potential in backyard, semi-intensive 265 and intensive production sites. Good production practices are not considered by all pro- 266 ducers from prevention, treatment and slaughter, due to lack of maintenance of facilities 267 [52].” How does this all correlate to your work? Explain or remove from discussion
R In the present study, we observed that many countries use large amounts of antibiotics which are found in animal products of any animal species in different concentrations, which confirms the above mentioned.
Line 271: “Oxytetracy- 270 cline, florfenicol, sarafloxacin, erythromycin and sulfonamides are widely used in aqua- 271 culture and are therefore detected in aquatic samples [54]. “ is this what you saw in your analysis? Discuss your results!
R has been. discussed
Line 283: “4.3. Impact of antibiotic residues in water and soil “ Please specify is waste water.
has been described.
Line 288: “In the United States (US), 70% of antibiotics were used in animal agriculture, which 288 is eight times the amount used in human medicine [58]; in 2013, US livestock producers 289 purchased 14,900 tons of antimicrobials [59]. Van Boeckel et al. [19] estimate that from 290 2010 to 2030, antibiotic use in food animal production will increase by 67% from 63,151 ± 291 1,560 tons to 105,596 ± 3,605 tons. “ how does this all correlate to your work? Explain or remove from discussion
R With the aforementioned background, such as the inappropriate use of antibiotics in animal production, antibiotic residues are excreted in faeces or urine and present in environmental matrices, such as soil, water and vegetation, which may cause risks to human health [80], hence the latest trends in antimicrobial resistance. For example, in the United States (US), 70% of antibiotics were used in animal production, which is eight times the amount used in human medicine [81]; in 2013, US livestock producers purchased 14,900 tonnes of antimicrobials [82] which represents a large amount of antibiotics potentially disposed of into the environment (aquatic waste and soil). Van Boeckel et al. [25] estimate that, between 2010 and 2030, the use of antibiotics in food animal production will increase by 67%, from 63,151 ± 1,560 tonnes to 105,596 ± 3,605 tonnes, which could increase the issue of antimicrobial resistance in future generations.
Line 311: “It has been shown through molecular markers that the main cause of antibiotic resi- 311 dues in the environment is due to livestock feces, as doses much higher than prescribed 312 are applied to the human population, as demonstrated in animal excrement in the United 313 States, where values exceeded 100 times more than those found in wastewater of anthro- 314 pogenic origin [70]. “ Does this match your results? Are the same antibiotics in water than in animals? Please Discuss
Line 316: “It is important to control the doses used in livestock production to avoid pathogen 316 resistance to antibiotics and water contamination, as wastewater in some areas is used as 317 an alternative for irrigation of agricultural fields. “ Add reference
- has been added
Mshana, S.E.; Sindato, C.; Matee, M.I.; Mboera, L.E.G. Antimicrobial Use and Resistance in Agriculture and Food Production Systems in Africa: A Systematic Review. Antibiotics. 2021, 10, 976. doi.org/10.3390/antibiotics10080976
Robertson Jane, Vlahović-Palčevski Vera, Iwamoto Kotoji, Högberg Liselotte Diaz, Godman Brian, Monnet Dominique L., Garner Sarah, Weist Klaus, Variations in the Consumption of Antimicrobial Medicines in the European Region, 2014–2018: Findings and Implications from ESAC-Net and WHO Europe. Frontiers in Pharmacology, 2021; 12 , 727 URL=https://www.frontiersin.org/article/10.3389/fphar.2021.639207 DOI=10.3389/fphar.2021.639207
Line 318: “According to a study by Xu et. al [71] in 318 urban rivers in Beijing, China, the sources of contamination are in places where large 319 amounts of antibiotics are used, such as hospitals.” You are looking at livestock not humans, How does this all correlate to your work? Explain or remove from discussion
R has been removed
Line 332: “Antibiotics can affect bacterial enzyme activity including dehydrogenases, phospha- 332 tases and ureases which are considered as indicators of soil biological activity. With the 333 presence of antibiotics in the environment this may favor the increase of pathogens or 334 parasites in soil and water. For example, the presence of antibiotic contaminants in water 335 favors the increase of toxic Cyanobacteria species that cause the phenomenon of eutroph- 336 ication in freshwater systems and with it possible risks to human health [74]. “Your study do not look at bacterial populations resistant to antibiotics, and although I find that topic fascinating, I do not think it is suitable to discus in depth in here.
R it has been deleted , thank you for you recomendation
Line 338: “With the development and improvement of laboratory analytical techniques, 338 changes to international and national legislation have been promoted. For example, the 339 European Commission [20,75], US Food and Drug Administration [76], and Codex Ali- 340 mentarius [77] have established maximum residue limits (MRLs) for VAs in foodstuffs 341 from animal origin [78]. Therefore, in recent years, rapid methods with the advantage of 342 easy performance, high sensitivity and high throughput are being proposed and used ex- 343 tensively. 344 “How does this all correlate to your work? Explain or remove from discussion
R has been removed
Line 347: “The use of antibiotics is defined 347 by the different local policies and legislation in each region according to the residual ef- 348 fects they have on the environment and their possible effects on human health” How can you conclude this if you have not looked at the differences between countries or comunities?
R has been removed. and. re write it
The main antibiotics used worldwide present in water residues, soil, and in animal pro-duction are sulfonamides, tetracyclines, quinolones, penicillin and cephalosporins. At present, there is still misuse of antibiotics in many countries, producers still do not want to stop using them as they believe that it would be impossible to sustain the current market demands with-out the use of antimicrobial or alternatives. We need to become aware that antibiotic contami-nation is a global problem, and we are challenged to reduce and improve their use.
Line 349: “At present, 349 although the use of antibiotics is being banned worldwide, producers still do not want to 350 stop using them as they mention that it would be impossible to sustain the current market 351 demands without large quantities of antimicrobial. : This is exactly what you said in your introduction, so you knew that before doing your study and is also not true. This is not a discussion. What about the residue in animal tissues and in environmental samples? nothing to conclude about that?
R has been rewrite. it as follows:
In recent years it has been demonstrated that antibiotic residues are present worldwide in wastewater, soil and animal production, the most common being sulphonamides, tetracyclines, quinolones, penicillins and cephalosporins. New international policies have limited their use as therapeutics, restricting their use as growth promoters in animal production. Intensive livestockproduction must change, as it would be impossible to sustain current market demands without the use of antimicrobials or friendlier alternatives, with a future decrease in antimicrobial resistance, so we are challenged to reduce their use.
Reviewer 2 Report
The topic of the publication is very timely and is part of the trend to reduce the use of antibiotics. The author's used correct research methodology and selection of articles for analysis. The authors accurately identified the risks associated with the excessive use of antibiotics and drew correct conclusions. I recommend the article for publication without amendments.
Author Response
Thank you very much for your approval, we have considered improve the manuscript according to the suggestion of reviewer 1, then you can see the. changes. in the manuscript colored in red
Reviewer 3 Report
The review by Robles Jimenez et al focuses on the use of antibiotics in farm animals. The manuscript is interesting, especially since it presents the links between the use of antibiotics and the traces left by them on meat or the environment. I have a few minor comments:
1. L. 30. Remove the "point" and make a sentence.
2. L. 37. I am not convinced that the use of antibiotics is being banned. Antibiotics can still be used to treat animals. It is the controversial use as a growth promoter that is increasingly banned.
3. Information on the 173 articles used should be compiled and presented as supplementary material.
4. Figure 1. Tetracyclines are not in the legend for Asia and Africa. Also, different sizes of fonts are used.
5. Figure 3. Why not put the graphs two by two as before?
6. L.248, 258 and 280. The names of bacteria should be in italics.
7. L. 327. How can fungi be affected by antibiotics?
Author Response
Dear editor. and reviewers, we are very grateful for your comments, which have substantially improved this work. The. study. has. been. reanalysed. according. to. the. suggestions. of. reviewer. 1,. as. we. had. the. data. but. did. not. know. how. to. present. them. This. new. version. has. taken. more. time. as. we. had. to. reanalyse. and. present. the. data. and. include. the. list. of. articles. (table. 1S). and. the. number. of. studies. by. continent,. environment. and. animal. species. which. ultimately. eliminated. the. few. data. we. had. for. equines. and. rabbits.
We hope that the reviewers' comments have been addressed and look forward to hearing from you if you have any comments.
The authors
The review by Robles Jimenez et al focuses on the use of antibiotics in farm animals. The manuscript is interesting, especially since it presents the links between the use of antibiotics and the traces left by them on meat or the environment. I have a few minor comments:
- L. 30. Remove the "point" and make a sentence.
R Has been attended - L. 37. I am not convinced that the use of antibiotics is being banned. Antibiotics can still be used to treat animals. It is the controversial use as a growth promoter that is increasingly banned.
- we are agree and has been change it
- Information on the 173 articles used should be compiled and presented as supplementary material.
R Has been attended , at the end. we consider 165 studies. with the recommendations of reviewer 1
see Supplemental table 1
- Figure 2. Tetracyclines are not in the legend for Asia and Africa. Also, different sizes of fonts are used.
R Has been attended, and changed, because. we had concentration of antibiotics in each study, then. we reanalyze. the. data, considering. with more powerfull our results and. discussion
- Figure 3. Why not put the graphs two by two as before?
R Has been attended, and change it
- L.248, 258 and 280. The names of bacteria should be in italics.
R Has been attended
- L. 327. How can fungi be affected by antibiotics?
You are. right and has been change it , as microorganism
Round 2
Reviewer 1 Report
The Authors have done an outstanding amount of work reanalyzing and reformatting the manuscript. I think this piece of work is very close to reach publication standards.
Two minor concerns:
- All figures present very low resolution to the point that I can read any of the names.
- Discussion sub-header 4.3 is missing
Once this two things are corrected I will accept the manuscript for publication.
Congratulations for your work!
Author Response
Dear reviewer, thank you very much for your comments, please find attached the figures with better resolution and annex 4.3. Impact of antibiotic residues in water and soil, we appreciate very much your comments which have substantially improved the manuscript and we hope it will be of interest to the general public and the scientific community.